# Maximum Likelihood Curved Surface Estimation of Multi-Baseline InSAR for DEM Generation in Mountainous Environments

**DOI:** 10.3390/s25113371

**Published:** 2025-05-27

**Authors:** Dehao Liang, Yugang Tian, Xinbo Liu, Haijing Ren, Huifan Liu

**Affiliations:** 1Hubei Key Laboratory of Intelligent Yangtze and Hydroelectric Science, China Yangtze Power Co., Ltd., Yichang 443000, China; cug_liang@126.com (D.L.); lxb@mail.bnu.edu.cn (X.L.); 05171733@cumt.edu.cn (H.R.); leontp@foxmail.com (H.L.); 2School of Geography and Information Engineering, China University of Geosciences, Wuhan 430074, China

**Keywords:** multi-baseline InSAR, maximum likelihood curved surface estimation, DEM generation, mountainous environments

## Abstract

Digital elevation model (DEM) generation using Interferometric Synthetic Aperture Radar (InSAR) in mountainous environments encounters challenges including signal acquisition difficulties, decorrelation, and highly variable topography. To address these challenges, we propose a novel approach termed maximum likelihood curved surface estimation (MLCSE), utilizing multi-baseline InSAR to enhance DEM accuracy in mountainous regions. First, multi-baseline InSAR with Sentinel-1 images is employed to acquire more accurate interferometric phases. Second, two strategies are implemented to improve maximum likelihood elevation estimation, which is particularly susceptible to topographic relief and decorrelation. These strategies include replacing fixed neighborhood size with adaptive neighborhood size selection and estimating parameters of the maximum likelihood local curved surface. Finally, the mean error of the MLCSE DEM results and the proportion of errors less than 10 m are 7.89 m and 70.32%, respectively. The results demonstrate that MLCSE surpasses other InSAR methods, achieving higher elevation estimation accuracy. MLCSE exhibits stable performance across the study areas, reducing elevation errors in hilly, mountainous, and alpine regions. Additionally, hydrological analysis of the elevation results reveals that MLCSE, using the adaptive neighborhood size selection strategy, outperforms other methods in both visual inspection and quantitative comparisons. Moreover, the elevation accuracy achieved by MLCSE meets the standards of the American DTED-2, the Level 2 standard of the 1:50,000 DEM (Mountain), and the Level 1 standard of the 1:50,000 DEM (alpine region) for spatial resolution and height accuracy.

## 1. Introduction

The digital elevation model (DEM) constitutes a three-dimensional model of the terrain’s surface [1]. High-precision DEMs are crucial components of national spatial data infrastructures, widely utilized in eco-environmental assessments, sustainability planning, and national security and military operations [2]. Unfortunately, such DEMs are not readily available in complex mountainous regions. This is primarily because traditional surveying and mapping technologies struggle to acquire DEMs across large areas with rugged terrain, inhospitable environments, and challenging weather conditions [3].

Over the past few decades, Interferometric Synthetic Aperture Radar (InSAR) technology has been extensively utilized for large-scale topographic generation tasks due to its ability to acquire ground information in all weather and lighting conditions and its ability to detect terrain undulations with millimeter-level precision [4,5]. However, rugged terrain in mountainous regions presents three major challenges to traditional single baseline InSAR techniques for DEM generation. First, the side-looking imaging mode of SAR creates extensive shadow and layover areas, leading to phase signal loss. Second, significant topographic relief on slopes can induce phase jumps, leading to errors in the phase unwrapping process. Third, mountainous regions are often densely forested, leading to low coherence and making the interferometric phase more vulnerable to noise. Thus, current single-baseline InSAR technologies for DEM generation require further improvements to address these challenges in complex mountainous environments.

To overcome these issues, multi-baseline InSAR technology, which combines multiple interferograms from different frequency sensors and spatial baselines, has been proposed to estimate elevation [6,7,8], eliminating the restrictive phase continuity assumption while enhancing phase reliability through multi-source redundancy. Existing methodologies are broadly categorized into non-parametric and parametric approaches [9]. Non-parametric methods resolve phase ambiguities through geometric consistency analysis of multi-baseline interferograms, whereas parametric approaches model elevation as a hidden stochastic variable and optimize its estimation via maximum likelihood principles.

In the non-parametric approach, phase ambiguity is resolved by analyzing the relationships among multi-baseline interferograms, with typical methods including the Chinese remainder theorem (CRT) [10,11] and clustering methods [12,13,14]. In [15], the CRT method and the single-baseline phase unwrapping approach are combined to generate the Two-Stage Programming Approach (TSPA). However, these methods are highly sensitive to the noisy phase, particularly in complex mountainous regions where decorrelation and abrupt terrain-induced phase variations occur.

In the parametric approach, elevation is treated as a parameter to be estimated, and the elevation information is derived by utilizing the probability density function associated with the interferometric phases from multi-baseline InSAR. The parametric method, based on the principle of maximum likelihood (ML) estimation [16,17,18,19], is designed for DEM generation in discontinuous terrain and is applicable only in scenarios with a high signal-to-noise ratio and sufficient baseline interferograms. However, it estimates the elevation independently pixel by pixel, leading to discontinuous height variations and numerous irregular spikes, which significantly affect the accuracy of elevation estimation. Many methods focus on incorporating contextual information to impose deterministic constraints among neighboring pixels [20,21,22,23,24] and employing the plane model [25]. However, the plane model assumes the linearity of the terrain within a local range, an assumption that often does not hold in complex mountainous regions where violent elevation changes occur, such as hills and mountains. Furthermore, these methods typically employ fixed neighborhood sizes to extract contextual information, which may fail to adequately account for variations across diverse topographic features. An undersized neighborhood often induces severe decorrelation in densely vegetated mountainous regions, resulting in degraded phase signals and biased elevation estimates. Conversely, an oversized neighborhood integrates phase information from spatially heterogeneous regions, thereby causing oversmoothing of local terrain details. To sum up, the methods discussed above struggle to effectively utilize valid interferometric phases and accurately reconstruct terrain details in complex mountainous regions with significant topographic variation and low overall coherence [19].

In DEM generation using multi-baseline InSAR for mountainous regions, it is crucial to emphasize the accuracy of surface relief representation and decorrelation handling. To improve topographic detail restoration and mitigate the effects of decorrelation in mountainous regions, this study employs multi-baseline InSAR data, focusing on the maximum likelihood curved surface estimation (MLCSE) approach with two key innovations. The first one is coherence-driven adaptive neighborhoods. Unlike fixed-size windows, our method dynamically adjusts the neighborhood size based on local coherence. The second one is local curved surface modeling. We replaced the plane model with a curved surface model integrated into the ML framework, enabling nonlinear terrain elevation fitting, which is better suited for mountainous scenarios.

To enhance the reliability of MLCSE, this paper employs multi-baseline C-band Sentinel-1 data, preprocessed for interferometry. The remainder of this paper is organized as follows: Section 2 introduces the principles of maximum likelihood local curved surface estimation with adaptive neighborhood sizes. Section 3 outlines the improved processing flow. Section 4 details and analyzes the experimental results. Section 5 discusses these results, and finally, the conclusions are presented.

## 2. Methods

In multi-baseline InSAR, maximum likelihood elevation estimation treats the target height, h, as a parameter of the probability distribution of the interferometric phase, denoted as $F_{mb}\left(\phi |h\right)$. For a single interferometric phase, the probability density function is periodic. When phases from multiple baselines are combined, the multiplication of the likelihood function resolves multiple global maxima at each frequency, allowing the maximum likelihood principle to find the optimal solution for the target elevation [26]. Therefore, the multi-baseline InSAR maximum likelihood elevation estimation is expressed by (1), (2), and (3):(1)$$h_{ML}=arg_{h}\max F_{mb}(\phi |h)$$(2)$$F_{mb}(\phi |h)=\prod_{n=1}^{N}f(\varphi_{n}|h)$$(3)$$f(\varphi_{n}|h)=\frac{1}{2\pi }\cdot \frac{1-|\gamma_{n}{|}^{2}}{1-|\gamma_{n}{|}^{2}\cos^{2}(\varphi_{n}-dis_{n}\frac{4\pi }{\lambda }h)}\cdot \left\{1+\frac{|\gamma_{n}|\cos(\varphi_{n}-dis_{n}\frac{4\pi }{\lambda }h)\cos^{-1}\left[-|\gamma_{n}|\cos(\varphi_{n}-dis_{n}\frac{4\pi }{\lambda }h)\right]}{{\left[1-|\gamma_{n}{|}^{2}\cos^{2}(\varphi_{n}-dis_{n}\frac{4\pi }{\lambda }h)\right]}^{\frac{1}{2}}}\right\}$$
where $\phi =[\varphi_{1}\left(i,j\right)\;\varphi_{2}\left(i,j\right)\cdot \cdot \cdot \varphi_{n}\left(i,j\right)]$ is the measured interferometric phase data vector, $\left(i,j\right)$ are the discrete range and azimuth coordinates, N is the number of baselines, $\varphi_{n}$ is the *n*th interferometric phase, h is the terrain height, $f\left(\varphi_{n}|h\right)$ is the height likelihood function for the *n*th interferogram, $\gamma_{n}$ is the *n*th coherence coefficient, λ is the wavelength corresponding the working frequency, and dis is a parameter related to the baseline of the InSAR system [26].

### 2.1. Maximum Likelihood Curved Surface Estimation

The maximum likelihood method estimates the elevation of each pixel by integrating phase and coherence information from multiple baselines, thereby generating the terrain for a specific region. In mountainous regions, numerous ground points often experience decorrelation and significant phase deviations, resulting in inaccuracies in elevation values derived from maximum likelihood estimation. To mitigate this issue, pixels within a local neighborhood are treated as a cluster to estimate the elevation of the central pixel, thereby improving estimation accuracy.

At the image location, $\left(i,\;j\right)$, the central pixel and its surrounding pixels can be considered as a cluster named $N_{ij}$, typically with a size of either 3 × 3 or 5 × 5. In mountainous areas with irregular slope variations, the terrain surface in a local neighborhood can be approximated as a curved surface. Within the cluster, the elevation of each pixel can be approximately expressed as illustrated in (4). The curved surface parameters are represented by the vector $x_{ij}={\left[a\left(i,j\right),b\left(i,j\right),c\left(i,j\right),d\left(i,j\right),e\left(i,j\right),f\left(i,j\right)\right]}^{T}$, while the measured phase data are denoted by the vector $\Theta_{ij}={\{\phi }^{T}\left(p,q\right),\;\left(p,q\right)\in \;N_{ij}\}$, where p and q are the positions of the pixels in the cluster. Thus, the estimation problem can be reformulated as shown in (5) and (6):(4)$$h(i,j)=a(i,j)p^{2}+b(i,j)q^{2}+c(i,j)pq+d(i,j)p+e(i,j)q+f$$(5)$$L(\theta_{ij}|x_{ij})=\prod_{p,q}F_{mb}(\phi (p,q)|x_{ij})=\prod_{p,q}\prod_{n=1}^{N}f(\varphi_{n}(p,q)|x_{ij})$$(6)$$\begin{array}{cc} f(\varphi_{n}(p,q)|x_{ij}) & =\frac{{\left(2\pi \right)}^{-1}(1-|\gamma_{n}{|}^{2})}{1-|\gamma_{n}{|}^{2}\;\cos^{2}\left(\varphi_{n}(p,q)-dis_{n}\frac{4\pi }{\lambda }(ap^{2}+bq^{2}+cpq+dp+eq+f)\right)}\\ & \times \left\{1+|\gamma_{n}|\cos\left(\varphi_{n}(p,q)-dis_{n}\frac{4\pi }{\lambda }(ap^{2}+bq^{2}+cpq+dp+eq+f)\right)\right.\\ & \times \cos^{-1}\left[-|\gamma_{n}|\cos\left(\varphi_{n}(p,q)-dis_{n}\frac{4\pi }{\lambda }(ap^{2}+bq^{2}+cpq+dp+eq+f)\right)\right]\\ & \left.÷{\left[1-|\gamma_{n}{|}^{2}\cos^{2}\left(\varphi_{n}(p,q)-dis_{n}\frac{4\pi }{\lambda }(ap^{2}+bq^{2}+cpq+dp+eq+f)\right)\right]}^{\frac{1}{2}}\right\} \end{array}$$
where $\gamma_{n}=\gamma_{n}(p,q)$, a=a(i,j), b=b(i,j), c=c(i,j), d=d(i,j), e=e(i,j), and f=f(i,j). Here, parameters a and b correspond to the curvatures along the *x*-axis and *y*-axis of the curved surface, respectively. By adjusting the value of parameters a and b, it becomes possible to model a range of complex topographic surfaces.

An illustration of curved surface fitting for various terrain types is shown in Figure 1. The details are as follows.

Canyon: When either a or b is 0 and the other parameter is positive, the curved surface will exhibit convex curvature along one axis, which can simulate canyon regions, per Figure 1a.

Ridge: When either a or b is 0 and the other parameter is negative, the curved surface will exhibit concave curvature along one axis, which can simulate ridge regions, per Figure 1b.

Basin: When both a and b are positive, the curved surface exhibits an upward convex curvature around the periphery, which can simulate a basin region with a central depression surrounded by higher elevation, per Figure 1c.

Peak: When both a and b are negative, the curved surface exhibits a downward convex curvature around the periphery, which can simulate a mountain peak region with higher elevation in the middle and lower elevation around it, per Figure 1d.

Saddle: When a and b have opposite signs, the curved surface exhibits convex curvature along one axis and concave curvature along the other axis, which can simulate a “saddle” region between two mountain peaks, per Figure 1e.

Parameter c represents the rotation and tilt of the surface; parameters d and e represent the gradients of the curved surface along the x- and y-axes, respectively; and f indicates the surface’s intercept with the z-axis, which provides an initial estimation for local curved surface elevation.

These six parameters collectively describe the details of the local curved surface, allowing for better estimation of terrain features and adaptation to regions under complex topography for DEM generation. The elevation value of the central pixel is obtained by estimating the curved surface parameters within the cluster based on the phase and coherence coefficients within the local domain.

### 2.2. Adaptive Neighbor Size Determination

To balance the restoration of geographic details with the acquisition of accurate phase information, the size of the local curved surface neighborhood should be determined by the coherence of the central pixel. When the central pixel’s coherence coefficient is high, a smaller neighborhood size can be used to minimize the influence of surrounding pixels and provide more detailed texture in the DEM. Conversely, when the coherence coefficient is low, a larger neighborhood size should be used to incorporate more valid phase observations, enabling accurate elevation estimation even in low-coherence regions.

The use of an adaptive neighborhood size preserves terrain details while generating high-precision DEMs in complex and noisy mountainous regions. In this study, a neighborhood size of 3 is applied to central pixels with high coherence, whereas a size of 5 is used for those with low coherence. After determining the neighborhood size, a threshold is applied to eliminate phase observations with significant deviations based on coherence coefficients, thereby enhancing the robustness of the elevation estimation.

## 3. Processing Flow of MLCSE

The processing flow of MLCSE utilizing multi-baseline InSAR can be systematically divided into three major stages. First, interferometric pair combinations, interferometric phases, and coherence coefficients are calculated. Second, elevation estimation is carried out using the principle of MLCSE. Finally, the estimated elevation map in SAR image coordinates is geocoded and projected into a geographic coordinate system, followed by accuracy validation. The complete flowchart of this approach is shown in Figure 2.

### 3.1. Interferometric Processing

The initial step involves selecting suitable SAR image pairs to generate interferograms after acquiring SAR SLC data over the same area from repeat-pass orbits. In mountainous regions with dense vegetation and complex elevation changes, SLC images were selected with the shortest temporal interval to minimize the effects of temporal decorrelation in the interferograms.

In this study, we utilized multi-baseline Sentinel-1 data, which has a 12-day revisit cycle. This short revisit cycle enables the acquisition of more valid phase observations. Additionally, 30 m resolution Copernicus Digital Elevation Model (COP-DEM) data were used as the prior DEM to register the SLC images. COP-DEM provides the highest absolute elevation and horizontal accuracy among all freely available DEMs. Its absolute vertical accuracy is less than 4 m, and relative vertical accuracy is less than 2 m at 90% confidence in all regions of the world. With up-to-date and detailed geographic features, COP-DEM offers accurate initial elevation values and search ranges.

The 12-day revisit cycle of the Sentinel-1 data was used as the shortest temporal baseline. Next, SAR images were registered for each interferometric pair; then, the interferometric phase and coherence coefficients were calculated. Goldstein filtering was subsequently applied to reduce speckle noise. Furthermore, tropospheric delay data from the Global Atmospheric Correction Observing System (GACOS) for the same period as the SAR images were used to correct for atmospheric delay phases.

### 3.2. Curved Surface Parameters Estimation

To estimate the elevation on a per-pixel basis, it is first necessary to define the size of the local neighborhood. The local neighborhood size is determined by calculating the average and standard deviation of coherence coefficients within the pixel set derived from multi-baseline interferograms. If the average coherence is high and the standard deviation is low, it indicates that the ground point maintains a high coherence state over time, warranting the selection of a neighborhood size of three. Conversely, if the average is low or the standard deviation is high, it suggests that the ground point is susceptible to decorrelation, in which case, a neighborhood size of five is selected. The thresholds for the average coherence coefficient and standard deviation are set at 0.4 and 0.1, respectively.

The subsequent step involves selecting valid phase observations from the local neighborhood, retaining only those with a coherence coefficient greater than 0.4. The COP-DEM data, projected into the SAR’s slant range coordinate system, provide the initial reference values for the subsequent maximum likelihood estimation. The elevation values of the DEM, encoded in radar coordinates, are subsequently utilized to compute the initial solution for the curved surface parameters using the least squares method. By integrating the selected phase and coherence coefficients, the initial likelihood probability for the curved surface parameters solution is computed in accordance with (5). The complete flowchart of curved surface parameters estimation is shown in Figure 3. The simulated annealing (SA) algorithm is then employed to optimize the curved surface parameters based on the maximum likelihood probability.

The simulated annealing (SA) algorithm is inspired by the process of metallurgical annealing, where the material is gradually cooled to reach a stable state. In the algorithm, several key parameters play a crucial role in guiding the optimization process. The temperature parameter, t, represents the degree of “randomness” in the search process, with a high temperature allowing for more exploration of the solution space. The initial annealing temperature, $t_{start}$, is the starting point, typically set high to encourage wide exploration of potential solutions. As the algorithm progresses, the termination temperature, $t_{end}$, is reached, marking the point at which the algorithm concludes, typically when the temperature falls below a predefined threshold. The cooling coefficient, c, determines the rate at which the temperature decreases during each iteration, controlling how quickly the algorithm “settles” into a solution. This gradual cooling process allows the algorithm to escape local optima and eventually converge to a global optimum, like how materials undergo a phase transition during physical annealing. The following outlines the implementation steps:

(1)Define the search range for the curved surface parameter solution, and specify the initial annealing temperature, $t_{start}$; termination temperature, $t_{end}$; and cooling coefficient, c.(2)Substitute the initial solution of the curved surface parameters into the likelihood function to calculate the initial probability, with the initial temperature, $t_{start}$, set as the current temperature, $t_{current}$.(3)Randomly select one of the six curved surface parameters, $x_{ij}$, from the current solution, applying the search step size to generate a new set of parameter values, $x_{ij}^{\prime }$. Then, compute the corresponding likelihood probability for the new solution.(4)If the likelihood probability of the new solution exceeds that of the current best solution, accept it as the new best solution and update the corresponding likelihood probability.(5)If the new likelihood probability is lower than the best, the solution may still be accepted with a probability determined by the current temperature, $t_{current}$. If accepted, the new solution replaces the current one, and the process returns to step (3) for the next iteration. Otherwise, the algorithm retains the previous solution and proceeds to the next iteration.(6)After a predefined number of iterations, reduce the temperature and repeat steps (3) through (5). This cycle continues until the global maximum likelihood probability and the corresponding optimal curved surface parameter solution are reached.

### 3.3. Geocoding and Validation

To ensure precise comparisons between DEMs derived from multi-baseline InSAR data and global datasets, the models are geocoded to the World Geodetic System 1984 (WGS84). This transformation facilitates both precision validation and terrain feature analysis. Both systematic and random errors introduced during the data generation process must be considered when evaluating DEM accuracy. The root mean square error (RMSE) is a common metric for assessing the alignment between actual values and estimates [27,28]. Mean error (ME) and standard deviation (STD) are also employed to quantify DEM errors [29,30,31,32,33,34,35,36,37,38], particularly when reference measurements, such as GPS points, demonstrate accuracy exceeding two orders of magnitude. As a derivative of elevation, slope parameters are more sensitive to non-systematic DEM elevation errors and can better reflect the terrain undulations in complex mountainous areas. Under the assumption of a normal distribution, specific formulas are used to calculate the ME, STD, and RMSE in (7), (8), and (9).(7)$$ME=\hat{\mu }=\frac{1}{N}\sum_{i=1}^{N}\left|{DEM}_{i}-{DEM}_{reference}\right|$$(8)$$STD=\hat{\sigma }=\sqrt{\frac{1}{N-1}\sum_{i=1}^{N}{\left({DEM}_{i}-\hat{\mu }\right)}^{2}}$$(9)$$RMSE=\sqrt{\frac{1}{N}\sum_{i=1}^{N}{\left({DEM}_{i}-{DEM}_{reference}\right)}^{2}}$$
where *N* is the number of samples.

Achieving a normal distribution of errors in InSAR-derived DEMs is uncommon. This can be attributed to various factors such as interpolation, filtering, layover, and shadow effects, especially in mountainous regions. A combination of graphical and statistical methods is used to evaluate the error distribution. Histograms are often used to facilitate visual inspection [39].

DEMs are commonly applied in practical fields, including terrain feature extraction and flood simulation. However, it is difficult to fully comprehend how well a DEM represents geographic details by merely analyzing ME, STD, and RMSE. In cases where DEM generation results exhibit similar error values, hydrological analysis can be utilized to extract drainage networks [34,38], which subsequently enables the delineation of ridges and canyons. Sketches of ridge and canyon lines provide a clear representation of the major relief structures in mountainous areas. These characteristics facilitate discussion regarding the functionality and efficiency of terrain features in estimating DEMs [40].

## 4. Experiments and Results

Xianyou County, located in Putian City, was chosen as the experimental area for implementing the MLCSE. Comparative experiments were conducted to evaluate the effectiveness of the adaptive local neighborhood size and the curved surface estimation module in improving the accuracy of elevation estimation.

### 4.1. Experimental Area and Data

The experimental area is located in Xianyou County, northwest of Putian City. The geographical location of the experimental area is shown in Figure 4. The region encompasses a diverse range of topographical features, including hills, mountains, ridges, basins, and canyons. The area under study is characterized by a complex mountainous landscape, marked by significant elevation differences and intricate topographical features. As detailed in Table 1, the DEM data indicate that the region is a complex mountainous area with significant elevation differences and intricate topography. Elevation ranges from a maximum of 1388.02 m to a minimum of 361.06 m, showing substantial variation in the terrain. The slope is generally steep, with an average of 26.41° and a maximum local slope of 78.4°, highlighting the terrain’s complexity. Roughness values range from 1 to 5.23, reflecting the surface’s irregularities and undulations. The relief reaches a maximum of 53.63 m, showing considerable vertical variation. The large standard deviation points to significant spatial variability in the surface undulations across the area. This variation in topography makes the region an ideal study area for multi-baseline InSAR DEM generation experiments.

The SAR interferometric data used in this experiment consisted of eight images acquired from the ascending-orbit Sentinel-1 mission. These data, acquired in winter and characterized by stable vegetation and the absence of snow cover, effectively reduce the risk of temporal decorrelation. In the experiment, eight images were combined into five interferometric pairs, employing the shortest temporal baseline. The temporal baselines of these pairs were 12 days, with spatial baselines ranging from −96.2061 m to 67.9097 m, indicating the inclusion of both long and short baselines. The height ambiguity for the interferograms ranged from 153.427 m to 1279.57 m, further reflecting the diversity in baseline lengths.

To enhance the signal-to-noise ratio, the SAR images were multi-looked with a 4:1 ratio in the range and azimuth directions. Additionally, the interferometric phase was filtered using Goldstein adaptive filtering, yielding a final spatial resolution of 20 m for the filtered interferogram. Figure 5 illustrates the spatiotemporal baseline distribution of the interferometric pairs, with each pair connected by a blue line. Figure 6 shows the interferograms of the Sentinel-1 data. The interferometric fringes are sparse in interferograms I and IV, with a short baseline, and become very dense in interferograms II, III, and V, with a long baseline. Table 2 presents detailed information on the master and slave image dates, temporal and spatial baselines, height ambiguity, coherence, and phase noise variance coefficient of the five interferograms.

In the experiment, a 30 m resolution COP-DEM was used as the prior DEM, which offers overall vertical accuracy better than 4 m. To validate the accuracy of the DEM reconstructed from multi-baseline InSAR data, a LiDAR-derived DEM obtained from the ShanHai Surveying and Mapping Company in Putian City was used as a reference. This reference DEM, originally at a resolution of 2 m, was downsampled to 20 m, providing 35,731 validation data for the accuracy assessment of the reconstructed DEM.

### 4.2. Experimental Results

The vertical height error of the COP-DEM is reported to be less than 4 m. As a result, the search range for central pixel elevation estimation was set to 8 m. To calculate the local curved surface likelihood probability, the COP-DEM was transformed into the radar coordinate system and used as the prior DEM. After determining the neighborhood size, the initial curved surface parameter values were extracted from the radar-coded COP-DEM. The reference curved surface parameters within this neighborhood were estimated using the least squares method, yielding the initial solution for the curved surface parameters. All InSAR DEMs were subsequently geocoded into the geographic coordinate system to present the final DEM generation results.

Figure 7 presents the DEM generation results from both single-baseline and multi-baseline InSAR. The phase unwrapping method used in single-baseline InSAR is the minimum cost flow method. The multi-baseline InSAR elevation reconstruction uses the MLE, MLPE, TSPA, and MLCSE methods. Figure 8 presents the slope results generated based on the DEM. The rows in Figure 7 and Figure 8 correspond to different regions. Excluding optical effects and the reference DEM, the first five columns display DEM generation results from the single-baseline InSAR, while the last three columns show results from the multi-baseline InSAR.

To quantitatively evaluate the elevation accuracy of the DEMs, all were geocoded into a geographic coordinate system with a spatial resolution of approximately 20 m. This evaluation aimed to validate the elevation improvements achieved through MLCSE. Figure 9 displays the height error maps for both single- and multi-baseline InSAR DEMs, illustrating the distribution and magnitude of height errors across the different methods. Figure 10 displays the slope error maps for both single- and multi-baseline InSAR DEMs. Additionally, Table 3 presents the statistical values derived from the height error maps for the corresponding InSAR DEMs shown in Figure 9, and Table 4 presents the statistical values derived from the slope error maps in Figure 10. This table effectively illustrates the statistical characteristics of the height error distribution.

## 5. Discussion

### 5.1. Comparative Analysis of the Single- and Multi-Baseline InSAR DEMs

When comparing DEMs generated from single-baseline and multi-baseline InSAR, significant differences in performance emerge. Single-baseline DEMs, estimated using interferograms I and IV, exhibit shorter baselines. This leads to low elevation sensitivity and a reduced ability to detect terrain variations, resulting in minimal topographic detail and substantial height errors. In contrast, interferograms II, III, and IV, which use longer baselines, provide improved elevation sensitivity and capture more geographic details. However, the rugged mountainous terrain and surface decorrelation introduce significant errors during the phase unwrapping process. The multi-baseline MLCSE method employed in this study effectively mitigates these challenges. In the Sentinel-1 experiment, the elevation ME, STD, and RMSE in the multi-baseline InSAR MLCSE DEMs are significantly lower than those in single-baseline DEMs. Additionally, a larger proportion of areas with height errors less than 10 m can be observed in the multi-baseline InSAR MLCSE DEMs, which also demonstrate a superior capacity for capturing geographic details. Thus, MLCSE, by mitigating phase unwrapping errors, is proved to be more effective for DEM generation in mountainous regions characterized by high noise and significant terrain relief.

### 5.2. Comparative Analysis of Multi-Baseline InSAR ML Algorithm

A comparative analysis of the error statistics from various multi-baseline InSAR DEM estimation algorithms is shown in Table 3 and Table 4. It reveals that MLE exhibits the largest ME and the lowest percentage of height errors less than 10 m and slope errors less than 5 degrees. MLPE performs slightly better than MLE. The performance of TSPA demonstrates superior accuracy and efficiency compared to both MLE and MLPE. MLCSE demonstrates the lowest ME, STD, and RMSE, as well as the highest percentage of height errors less than 10 m and slope errors less than 5 degrees.

To illustrate the differences between the various multi-baseline InSAR elevation estimation models more intuitively, the reference DEM was converted into a slope map, and pixels were classified into distinct terrain types. Slopes ranging from 0° to 2° were classified as plains, those from 2° to 6° as hills, slopes from 6° to 25° as mountains, and slopes exceeding 25° as alpine regions. Histograms of height errors for the different slope categories are shown in Figure 11, and the corresponding statistical values are presented in Table 5.

Compared to MLE, MLPE, and TSPA, the MLCSE yields DEM generation results with smaller errors, as shown in Figure 11a–d. It is evident that as the terrain becomes more complex, the accuracy of DEM generation decreases for all four multi-baseline InSAR algorithms. As the slope increases, the errors for MLE, MLPE, and TSPA increase significantly, whereas the error increase for MLCSE is comparatively gradual. In alpine regions, the MEs for MLE and MLPE exceed 11 m and for TSPA exceed 9 m, as illustrated in Figure 11q–s. This occurs because these algorithms do not filter the interferometric phase, resulting in highly biased phase information affected by noise still being included in the maximum likelihood estimation. Consequently, this leads to a final elevation estimate with significant errors. Furthermore, MLE estimates elevation on a pixel-by-pixel basis; the generated terrain often exhibits “spikes” in many areas, resulting in poor terrain continuity. The MLPE method estimates parameters within a local plane, resulting in smoother and more natural terrain. However, in regions with significant topographic relief, the plane model inadequately fits the frequently changing slopes. This inadequacy results in a poor representation of complex features, such as ridges and canyons. TSPA is susceptible to discontinuous phases caused by decorrelation, resulting in significant errors during the first stage of minimum ambiguity gradient estimation. These errors propagate in subsequent stages, making the method unsuitable in alpine regions with low coherence.

Across various slope categories, the MLCSE consistently demonstrates superior accuracy and stability, indicating that the curved surface model exhibits greater adaptability and precision in capturing local terrain variations, as shown in Figure 11d,h,l,p,t. The MLCSE maintains overall terrain continuity while accurately representing detailed features in complex terrain.

The MLCSE algorithm outperforms MLE, MLPE, and TSPA in complex terrains by employing a more flexible and precise curved surface model. This capability enables the algorithm to better accommodate local topographic variations, suppress noise, and accurately convey detailed features in complex landscapes. Therefore, in elevation estimation tasks, especially in regions with complex topography, the advantages of the MLCSE algorithm are especially pronounced.

### 5.3. Comparative Analysis of Adaptive Neighborhood Size in MLCSE

To investigate the impact of varying neighborhood sizes on DEM generation accuracy in MLCSE, curved surface fitting was conducted using fixed 3 × 3, fixed 5 × 5, and adaptive neighborhood sizes. The error estimation results are presented in Table 6, indicating minimal differences in the ME, STD, and RMSE values among the three methods.

To illustrate more intuitively how varying neighborhood sizes affect the simulation of local terrain details, prominent ridges and valleys were extracted in five distinct regions. The results for these ridge and valley profiles generated under different neighborhood sizes were verified by plotting cross-sectional profiles. For the line maps, the ridges and valleys were manually delineated based on the reference DEM. The comparison results are illustrated in Figure 12. In low-coherence areas, the terrain generated using the 5 × 5 neighborhood size (green line) and the adaptive neighborhood size (red line) strategy is more reliable, demonstrating closer agreement with the reference data (blue line), as demonstrated by the ridges in A3 and A4 (as shown in Figure 12: A3-Ridge1, A4-Ridge1, and A4-Ridge2). However, in high-coherence areas, the terrain data generated under the 3 × 3 neighborhood size and the adaptive neighborhood size strategy are more accurate for all ridges and canyons. The terrain line results generated under the 3 × 3 neighborhood size are smoother, while those produced under the 5 × 5 neighborhood size exhibit greater fluctuations, as observed in the ridge and canyons in A3 and A4 (as shown in Figure 12: A3-Canyon, A4-Ridge2, and A4-Canyon). This occurs because in regions with complex terrain, an excessively large neighborhood size may blend different features together, leading to erroneous extractions. Consequently, it becomes challenging to accurately capture small structures, such as ridges and canyons, resulting in incomplete and inaccurate extractions. Conversely, a neighborhood size that is too small becomes more sensitive to noise in low-coherence areas, resulting in interference and the erroneous extraction of terrain features. The adaptive neighborhood size strategy dynamically adjusts the neighborhood size based on local coherence. In high-coherence areas, the adaptive strategy selects a smaller neighborhood to capture local details more precisely. In low-coherence areas, the adaptive strategy selects a larger neighborhood to gather more effective interferometric phases, thereby suppressing noise effectively. This approach mitigates extraction errors caused by local noise and ensures the completeness of ridges and canyons.

MLCSE with an adaptive neighborhood size strategy, by dynamically adjusting neighborhood sizes, better balances detail capture and noise suppression, rendering it superior in extracting ridges and canyons in complex terrains. In contrast, fixed-neighborhood-size strategies, due to their inflexibility, either fail to capture details or introduce excessive noise in various regions, resulting in incomplete and erroneous extraction outcomes. MLCSE with an adaptive local neighborhood size is more suitable for terrain estimation tasks in mountainous regions characterized by complex terrain.

### 5.4. Analysis of Adaptive Multi-Baseline InSAR MLCSE DEM and COP-DEM

A comparative analysis was performed between the multi-baseline InSAR MLCSE DEM and the COP-DEM. Incorporating the COP-DEM as prior information significantly enhanced the quality of the MLCSE DEM, reducing errors and improving the overall accuracy of the final DEM. The multi-baseline InSAR MLCSE DEM enhances terrain boundary delineation compared to the COP-DEM, as shown in Figure 13. The contours of the ridgelines and canyon lines are more pronounced (particularly the third column in Figure 13). As presented in Table 7, the statistical values of height errors indicate that the ME and RMSE of MLCSE are lower than those of the COP-DEM. Furthermore, the spatial resolution of the InSAR MLCSE DEM is 20 m, compared to 30 m for the COP-DEM, allowing for a clearer representation of surface undulations. Consequently, the MLCSE DEM, derived from the COP-DEM as prior information, surpasses the COP-DEM in both resolution and terrain accuracy. Thus, the multi-baseline MLCSE DEM can serve as an optimization and enhancement of the COP-DEM.

According to the Digital Terrain Elevation Data (DTED) standards established by the National Geospatial-Intelligence Agency (NGA) of the United States, the multi-baseline InSAR DEM generated in this study meets the DTED-2 standard for spatial resolution and elevation accuracy, as indicated by the height error statistics in Table 7 [41]. Similarly, in reference to the 1:50,000 DEM standard for China published by the National Administration of Surveying, Mapping, and Geoinformation of China, the elevation accuracy of the multi-baseline InSAR DEM in mountainous regions meets the Level 2 standard of the 1:50,000 DEM (Mountain) and the Level 1 standard of the 1:50,000 DEM (alpine region) [42].

## 6. Conclusions

In this paper, we present a novel multi-baseline InSAR elevation generation method called maximum likelihood curved surface estimation (MLCSE). MLCSE effectively utilizes the interferometric phase values from multi-baseline InSAR through maximum likelihood estimation. By first establishing a curved surface estimation model, the method restores intricate topographic details. Subsequently, an adaptive local neighborhood size is implemented to suppress phase noise and enhance the robustness of the MLCSE. Finally, the effectiveness of this method is validated through DEM generation in the Xianyou region of Putian City using Sentinel-1 data. The main findings are summarized as follows:(1)Multi-baseline InSAR circumvents the phase unwrapping process inherent in single-baseline InSAR, enabling effective use of interferometric phase information, even in low-coherence areas, resulting in more accurate DEM reconstruction in complex mountainous regions.(2)The mean error of MLCSE DEM results and the proportion of errors less than 10 m are 7.89 m and 70.32%, respectively. The MLCSE outperforms other multi-baseline InSAR terrain estimation methods, especially in steep mountainous regions. The MLCSE more effectively utilizes local terrain information by employing a curved surface model that adapts to topographic variations and can recover terrain details in regions with significant elevation changes.(3)The adaptive neighborhood size is determined by the principle of employing a smaller window in high-coherence, low-noise areas and a larger window in low-coherence, high-noise areas to incorporate accurate phase information through curved surface parameter estimation. This strategy not only preserves terrain details but also mitigates noise, resulting in higher accuracy in DEM generation using an adaptive size compared to fixed sizes.(4)DEM generation using MLCSE demonstrates significant advantages in both spatial resolution and accuracy compared to the COP-DEM. Consequently, MLCSE represents an advancement over historical low-resolution DEM versions. Experiments also demonstrate that multi-baseline InSAR DEM generation from Sentinel-1 datasets using MLCSE meets the American DTED-2 standard, the Level 2 standard of the 1:50,000 DEM (Mountain), and the Level 1 standard of the 1:50,000 DEM (alpine region) for spatial resolution and height accuracy.

In summary, MLCSE offers a promising solution for DEM generation in mountainous environments characterized by significant terrain relief. In the future, the proposed multi-baseline InSAR MLCSE can be tested and improved further with more spaceborne datasets.

## Figures and Tables

**Figure 1 sensors-25-03371-f001:**
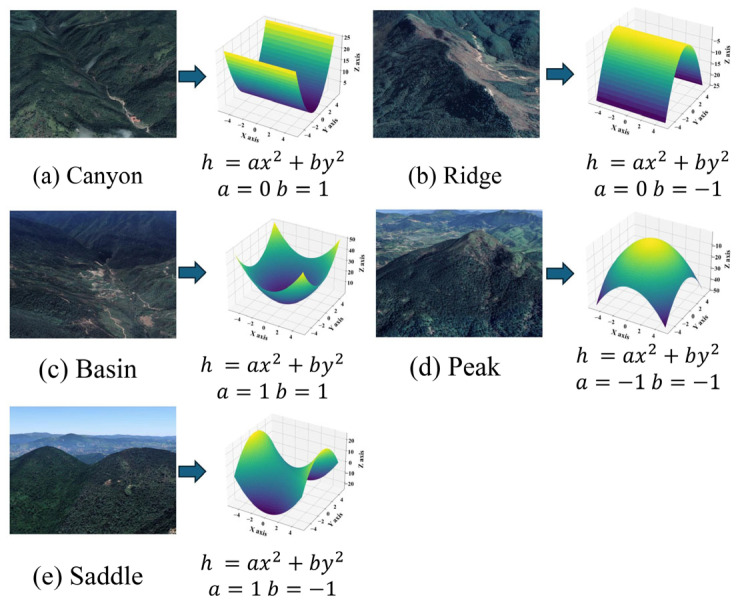
Illustration of surface fitting for various terrain types.

**Figure 2 sensors-25-03371-f002:**
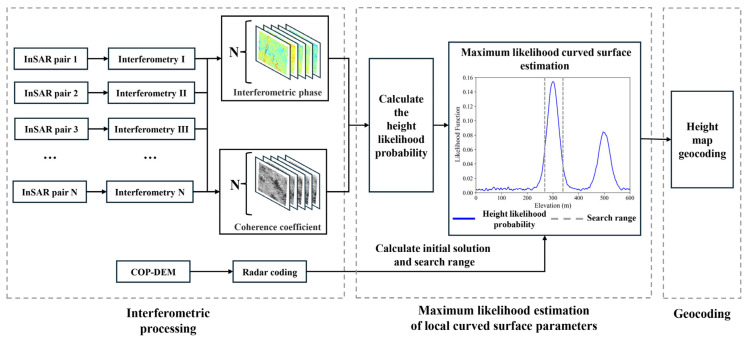
Flowchart for MLCSE DEM generation.

**Figure 3 sensors-25-03371-f003:**
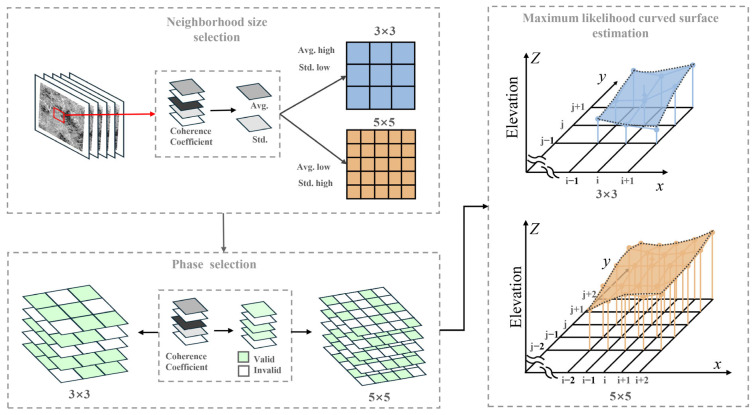
Flowchart for curved surface parameters estimation.

**Figure 4 sensors-25-03371-f004:**
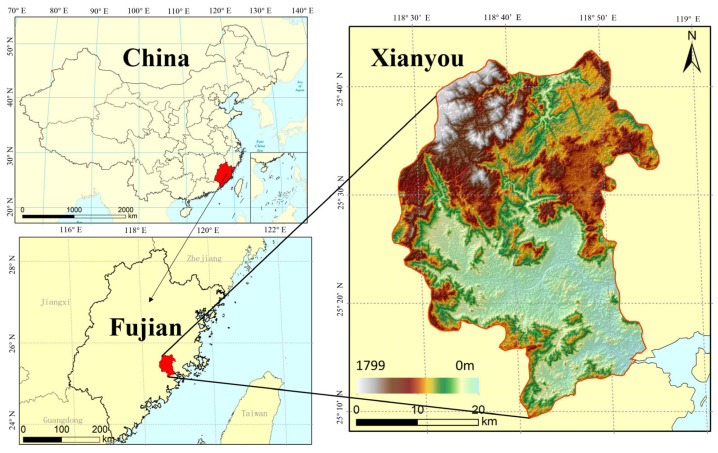
Study area.

**Figure 5 sensors-25-03371-f005:**
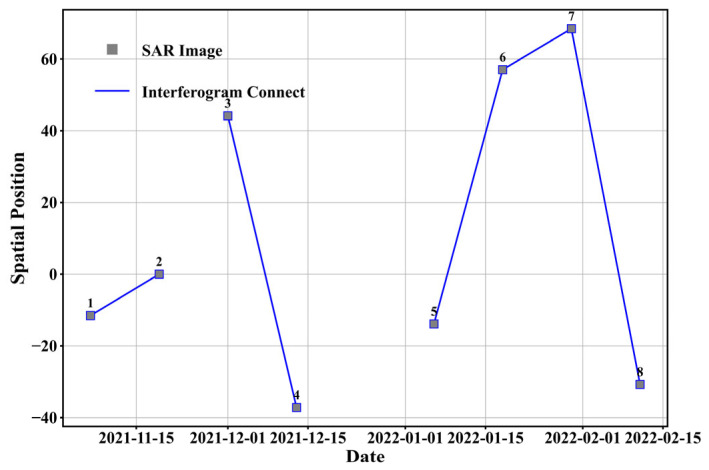
Spatial–temporal baseline distribution of Sentinel-1 interferometric pairs, with images forming pairs connected by blue lines. The serial number represents different SAR images.

**Figure 6 sensors-25-03371-f006:**
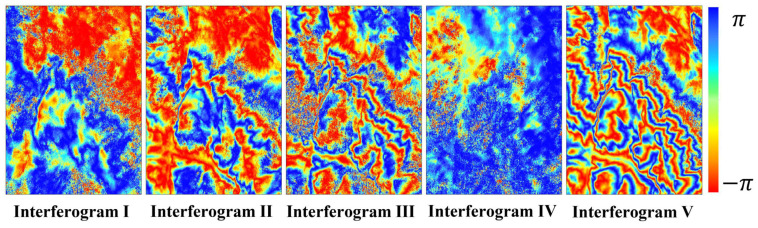
Wrapped phases of the multi-baseline SAR interferometric pairs used for DEM estimation.

**Figure 7 sensors-25-03371-f007:**
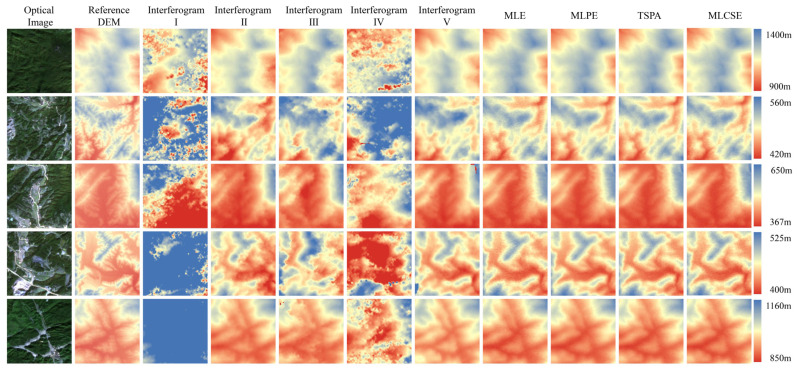
Height map under different single/multi-baseline InSAR algorithms.

**Figure 8 sensors-25-03371-f008:**
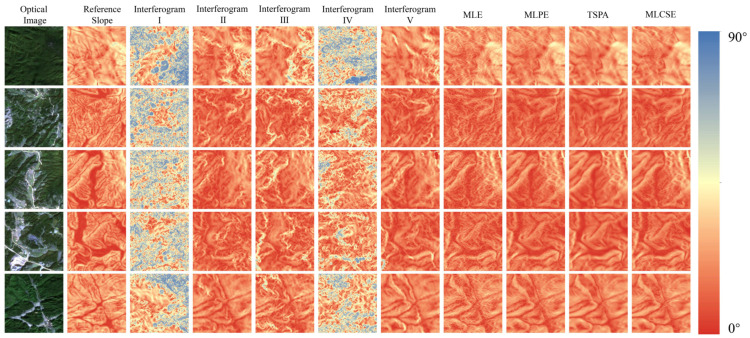
Slope map under different single/multi-baseline InSAR algorithms.

**Figure 9 sensors-25-03371-f009:**
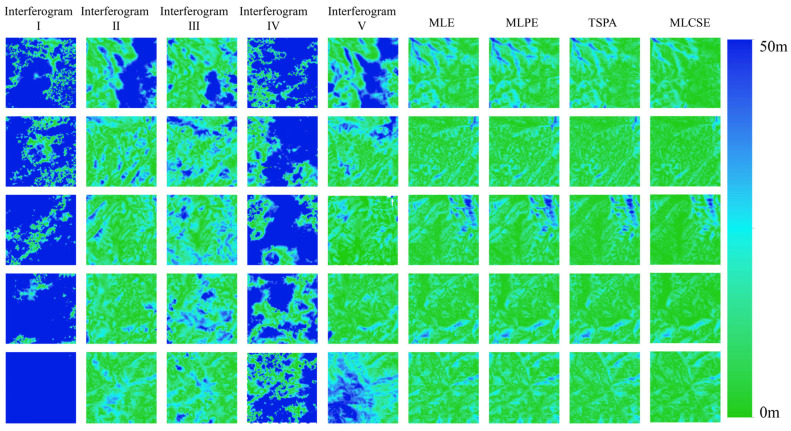
Height error map under different single/multi-baseline InSAR algorithms.

**Figure 10 sensors-25-03371-f010:**
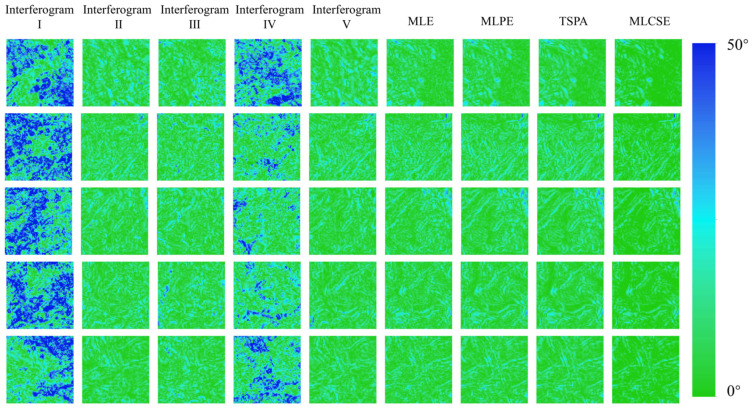
Slope error map under different single/multi-baseline InSAR algorithms.

**Figure 11 sensors-25-03371-f011:**
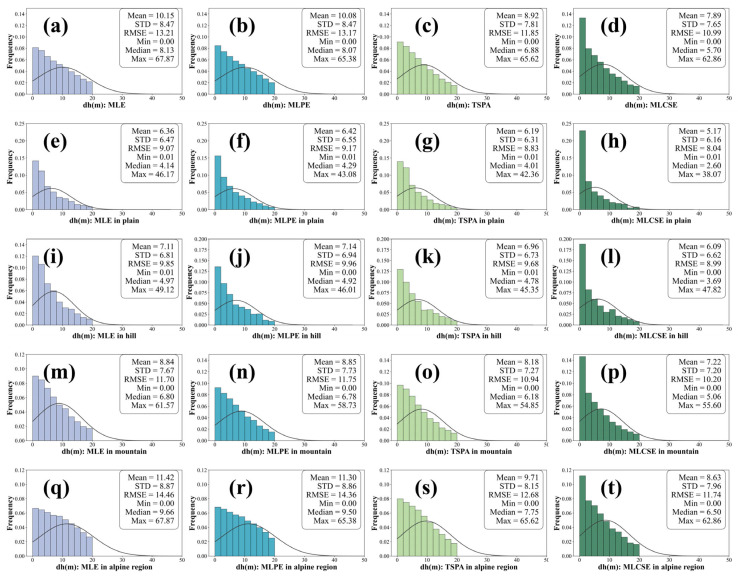
Histograms with height error statistics of multi-baseline InSAR DEMs algorithms in various slope categories. Subfigure (**a**–**d**) show the histograms with total height errors for MLE, MLPE, TSPA, and MLCSE, respectively. Subfigure (**e**–**h**) show the histograms with height errors in plain for MLE, MLPE, TSPA, and MLCSE, respectively. Subfigure (**i**–**l**) show the histograms with height errors in hill for MLE, MLPE, TSPA, and MLCSE, respectively. Subfigure (**m**–**p**) show the histograms with height errors in mountain for MLE, MLPE, TSPA, and MLCSE, respectively. Subfigure (**q**–**t**) show the histograms with height errors in alpine region for MLE, MLPE, TSPA, and MLCSE, respectively.

**Figure 12 sensors-25-03371-f012:**
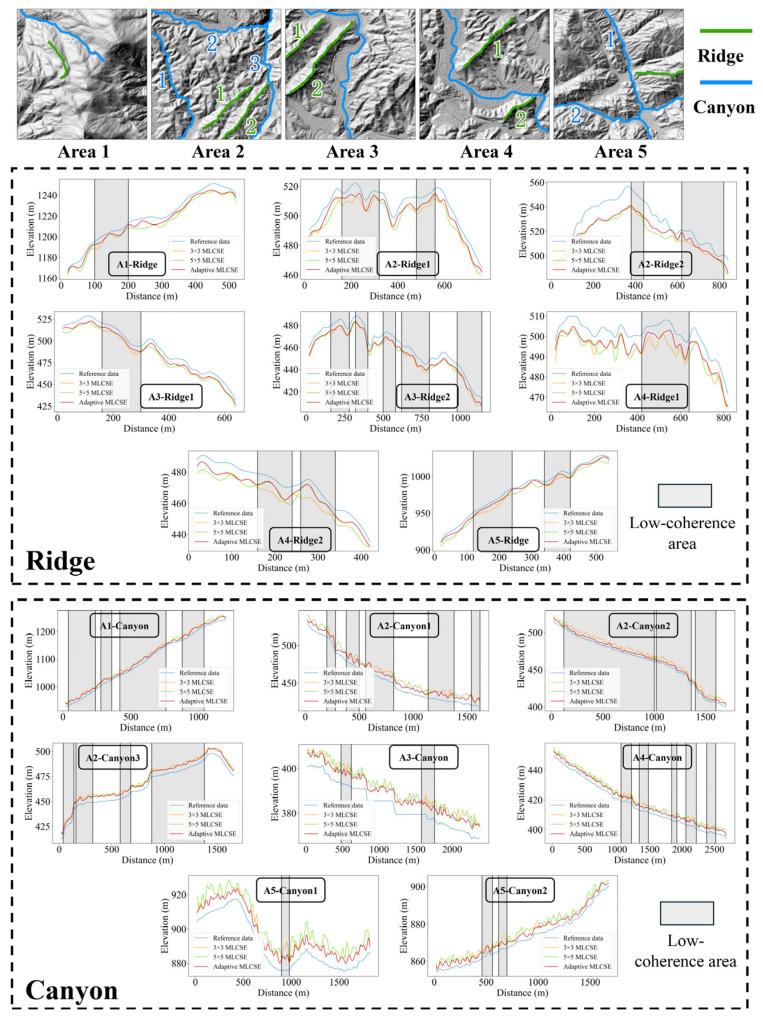
Ridge and canyon prediction using MLCSE with different neighborhood sizes.

**Figure 13 sensors-25-03371-f013:**
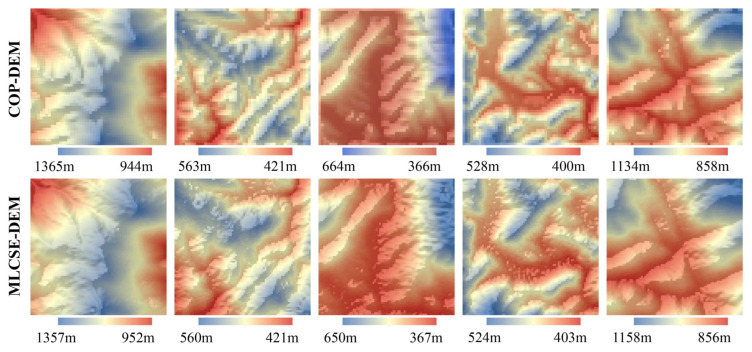
Wrapped phases of the multi-source SAR interferometric pairs used for DEM estimation.

**Table 1 sensors-25-03371-t001:** Topographical data statistics of the study area.

	Elevation (m)	Slope (°)	Roughness	Relief (m)
Maximum	1388.02	78.40	5.23	53.63
Mean	698.67	26.41	1.16	11.52
Median	497.99	23.23	1.21	10.76
Minimum	361.06	0.00	1.00	0.00
Std Dev	313.52	13.36	0.16	5.46

**Table 2 sensors-25-03371-t002:** Parameters for Sentinel-1 interferometric pairs.

Interferogram	Ⅰ	Ⅱ	Ⅲ	Ⅳ	Ⅴ
Master image date	7 November 2021	1 December 2021	6 January 2022	18 January 2022	30 January 2022
Slave image date	19 November 2021	13 December 2021	18 January 2022	30 January 2022	11 February 2022
Temporal baseline (days)	12	12	12	12	12
Normal baseline (m)	11.5	−78.7	67.9	12.1	96.2
Height ambiguity(m)	1280	187	217	1220	153
Mean coherence coefficient	0.47	0.50	0.45	0.50	0.51

**Table 3 sensors-25-03371-t003:** Statistical values of height errors of single/multi-baseline InSAR DEMs.

	ME	STD	RMSE	Absolute Value < 10 m
Interferogram Ⅰ DEM	147.04	131.72	196.50	4.76%
Interferogram Ⅱ DEM	16.35	17.81	23.73	46.30%
Interferogram Ⅲ DEM	17.75	14.04	22.24	32.53%
Interferogram Ⅳ DEM	58.41	53.88	73.07	11.68%
Interferogram Ⅴ DEM	17.86	16.50	23.95	39.50%
MLE DEM	10.15	8.47	13.21	58.43%
MLPE DEM	10.08	8.47	13.17	58.85%
TSPA DEM	8.92	7.81	11.85	65.56%
**ML** **C** **SE DEM**	**7.89**	**7.65**	**10.99**	**70.32%**

**Table 4 sensors-25-03371-t004:** Statistical values of slope errors of single/multi-baseline InSAR DEMs.

	ME	STD	RMSE	Absolute Value < 5°
Interferogram Ⅰ DEM	28.33	18.01	33.49	10.24%
Interferogram Ⅱ DEM	7.78	5.96	9.80	39.50%
Interferogram Ⅲ DEM	8.46	6.81	10.86	37.75%
Interferogram Ⅳ DEM	18.36	14.47	23.37	18.93%
Interferogram Ⅴ DEM	7.50	5.89	9.54	41.42%
MLE DEM	6.97	5.65	8.98	45.49%
MLPE DEM	6.93	5.61	8.93	45.89%
TSPA DEM	6.64	5.53	8.64	48.19%
**ML** **C** **SE DEM**	**5.67**	**5.29**	**7.76**	**56.57%**

**Table 5 sensors-25-03371-t005:** Statistical values of height errors of multi-baseline InSAR DEM algorithms in various slope categories.

	Terrain Types	ME	STD	RMSE	Absolute Value < 10 m
MLE	Plain	6.36	6.47	9.07	78.36%
	Hill	7.11	6.81	9.85	75.10%
	Mountain	8.84	7.67	11.70	65.53%
	Alpine Region	11.42	8.87	14.46	51.58%
MLPE	Plain	6.42	6.55	9.17	78.00%
	Hill	7.14	6.94	9.96	73.56%
	Mountain	8.85	7.73	11.75	65.83%
	Alpine Region	11.30	8.86	14.36	52.31%
TSPA	Plain	6.19	6.31	8.83	80.56%
	Hill	6.96	6.73	9.68	74.38%
	Mountain	8.18	7.27	10.94	69.64%
	Alpine Region	9.71	8.15	12.68	61.33%
MLCSE	Plain	**5.17**	**6.16**	**8.04**	**83.21%**
	Hill	**6.09**	**6.62**	**8.99**	**77.10%**
	Mountain	**7.22**	**7.20**	**10.20**	**73.95%**
	Alpine Region	**8.63**	**7.96**	**11.74**	**66.88%**

**Table 6 sensors-25-03371-t006:** Statistical values of height errors of MLCSE with different neighborhood sizes.

	ME	STD	RMSE	Absolute Value < 10 m
3 × 3 MLCSE	8.55	7.75	11.76	65.79%
5 × 5 MLCSE	8.70	7.81	11.63	67.01%
**Adaptive MLCSE**	**7.89**	**7.65**	**10.99**	**70.32%**

**Table 7 sensors-25-03371-t007:** Statistical values of height errors of MLCSE and COP-DEM.

	ME	STD	RMSE	Absolute Value < 10 m
COP-DEM	8.70	7.24	12.12	63.22%
**ML** **C** **SE DEM**	**7.89**	**7.65**	**10.99**	**70.32%**

## Data Availability

The original contributions presented in this study are included in the article. Further inquiries can be directed to the corresponding author.

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
