# Peer review of "Maximum Likelihood Curved Surface Estimation of Multi-Baseline InSAR for DEM Generation in Mountainous Environments"

_sensors, 2025, doi:10.3390/s25113371_

Round 1

Reviewer 1 Report

Comments and Suggestions for Authors

This paper focuses on the generation of digital elevation models in mountainous environments. In response to the challenges faced by traditional single-baseline InSAR technology, a maximum likelihood surface estimation (MLCSE) method is proposed. This method uses multi-baseline InSAR to obtain more accurate interferometric phases, and improves maximum likelihood elevation estimation through two strategies: adaptive neighborhood size selection and maximum likelihood local surface parameter estimation. The study used Xianyou County as the experimental area and conducted experiments using Sentinel-1 data. The results showed that MLCSE is superior to other InSAR methods in terms of elevation estimation accuracy and is stable in different terrain areas.This work could be accepted after minor revisions. Other questions were shown below.

  1. The parameter d in formula (2) is not clearly defined, and is only mentioned as “related to the InSAR system baseline”, lacking a specific physical meaning or calculation formula.
  2. The parameter descriptions (e.g., a=0, b=1) of the example terrains (e.g., “canyon”, “ridge”) in Figure 1 do not clearly explain their physical meanings in the captions.
  3. The latitude and longitude markings in Figure 4 do not clearly indicate the north latitude, south latitude, east longitude and west longitude.
  4. The colorbar in Figure 8 is redundant, so it is recommended to keep only one.
  5. When comparing the experimental results of different algorithms, the analysis is not comprehensive enough only through the statistical values ​​of height errors and the visualization of some areas. More indicators or methods can be added, such as the comparison of the extraction accuracy of terrain features, to more comprehensively evaluate the performance of the algorithms.
  6. When analyzing the impact of adaptive neighborhood size on DEM generation accuracy, only five areas were selected for verification. The sample size was small, and the universality of the conclusions may be affected. It is recommended to add more verification in different terrain areas.

Reviewer 2 Report

Comments and Suggestions for Authors

It would have been better if the abstract and conclusion provides at least 2-3 significant quantitative findings with their implications. It would also be better to discussed the intrinsic and extrinsic limitations of the proposed methods and provide immediate recommendations for future work. 

Reviewer 3 Report

Comments and Suggestions for Authors

See attached file with review

Round 2

Reviewer 3 Report

Comments and Suggestions for Authors

You didn't correct the typo in formula (6) in the line beginning with cos^(-1)(...). In the cosine argument, your height parameter appears in the denominator when it should be in the numerator. Additionally, you omitted the variable d here. Why does the wavelength have an index n? You're considering a multibaseline configuration, not multifrequency.
